# A functional link between the co-translational protein translocation pathway and the UPR

**Rachel Plumb†, Zai-Rong Zhang†, Suhila Appathurai, Malaiyalam Mariappan\***

Department of Cell Biology, Nanobiology Institute, Yale School of Medicine, West Haven, United States

**Abstract** Upon endoplasmic reticulum (ER) stress, the transmembrane endoribonuclease Ire1α performs mRNA cleavage reactions to increase the ER folding capacity. It is unclear how the low abundant Ire1α efficiently finds and cleaves the majority of mRNAs at the ER membrane. Here, we reveal that Ire1α forms a complex with the Sec61 translocon to cleave its mRNA substrates. We show that Ire1α's key substrate, *XBP1u* mRNA, is recruited to the Ire1α-Sec61 translocon complex through its nascent chain, which contains a pseudo-transmembrane domain to utilize the signal recognition particle (SRP)-mediated pathway. Depletion of SRP, the SRP receptor or the Sec61 translocon in cells leads to reduced Ire1α-mediated splicing of XBP1u mRNA. Furthermore, mutations in Ire1α that disrupt the Ire1α-Sec61 complex causes reduced Ire1α-mediated cleavage of ER-targeted mRNAs. Thus, our data suggest that the Unfolded Protein Response is coupled with the co-translational protein translocation pathway to maintain protein homeostasis in the ER during stress conditions.

**\*For correspondence:**
malaiyalam.mariappan@yale.edu

†These authors contributed equally to this work

**Competing interests:** The authors declare that no competing interests exist.

**Reviewing editor**: Reid Gilmore, University of Massachusetts Medical School, United States

## Introduction

Secretory and transmembrane proteins are essential for linking intracellular trafficking and extracellular environments and, in metazoans, play broad roles in all aspects of intracellular communication. These proteins contain either a signal sequence or transmembrane domain (TMD) that is co-translationally captured by the SRP and targeted to the endoplasmic reticulum (ER) membrane (*Akopian et al., 2013*). At the ER membrane, the Sec61 translocon facilitates insertion or translocation of these polypeptides into the ER membrane (*Park and Rapoport, 2012*). The ER contains a network of chaperones and enzymes to assist in folding proteins into their native conformations (*Braakman and Bulleid, 2011*). When the influx of nascent polypeptides exceeds the ER protein folding capacity, misfolded proteins accumulate in the ER and lead to ER stress. Under ER stress, signaling pathways, collectively termed the Unfolded Protein Response (UPR), are activated to restore ER homeostasis (*Walter and Ron, 2011*). The activation of the UPR leads to various cellular processes that include: transcriptional upregulation of UPR target genes, attenuation of translation, activation of ER associated degradation, and ER expansion. However, if ER homeostasis cannot be restored, the UPR induces cell death pathways to eliminate non-functional cells (*Walter and Ron, 2011*). The significance of the UPR is underscored by the fact that aberrations in UPR signaling can lead to a multitude of diseased states including neurological disorders, diabetes, and inflammatory disorders (*Wang and Kaufman, 2012*). In solid tumors, the UPR is constitutively activated as an adaptive response pathway for survival under adverse conditions, such as hypoxia (*Wang and Kaufman, 2012*).

Several ER transmembrane proteins act as ER stress sensors (*Walter and Ron, 2011*). The most ancient member of these, inositol-requiring enzyme 1 (Ire1), is conserved from yeast to mammals (*Cox et al., 1993*; *Mori et al., 1993*). Ire1 contains an ER luminal domain involved in sensing misfolded

**eLife digest** Proteins are made up of long chains of smaller building blocks called amino acids. To build this chain, a molecule called mRNA is 'translated' into the sequence of amino acids by a molecular machine called a ribosome. In order to work, the protein chain must then be folded into a complex shape. For many proteins, this happens inside a cell compartment called the endoplasmic reticulum.

Newly made proteins are guided to the endoplasmic reticulum by 'signal recognition particles', and then enter the endoplasmic reticulum through a channel protein called Sec61. If too many protein chains arrive at once, or they are folded too slowly, the accumulation of unfolded proteins can stress the endoplasmic reticulum. To fix this, cells trigger a process called the unfolded protein response.

In mammals, an enzyme called Ire1α detects when the endoplasmic reticulum is becoming stressed and responds by cleaving mRNA molecules. One particular target of Ire1α is the mRNA molecule that encodes a protein called XBP1, which can activate hundreds of genes to increase the size—and hence reduce the stress—of the endoplasmic reticulum. This protein is only made if a section of the mRNA molecule is removed from it; thus, by cleaving the mRNA, Ire1α enables the protein to be made. It remains unknown, however, how Ire1α finds and cleaves its mRNA targets.

Plumb, Zhang et al. identified the proteins that bind to Ire1α in human cells, and found that the Sec61 channel is one such protein. This interaction localizes Ire1α to the Sec61 channel in the endoplasmic reticulum membrane. The XBP1 protein is then brought to this channel by a signal recognition particle while it is still being translated—that is, when it is still attached to the ribosome and its mRNA molecule. Ire1α can then cleave the XBP1 mRNA. In cells that lack the signal recognition particle or the Sec61 channel protein, Ire1α cannot efficiently cleave the XBP1 mRNA molecule. In addition, if Ire1α is unable to interact with the channel protein, it does not efficiently cleave mRNA molecules at the endoplasmic reticulum membrane.

This work establishes a new link between the unfolded protein response and the pathway that brings new proteins to the endoplasmic reticulum membrane. It provides a basis for future studies examining the details of Ire1α signaling in mammals and, in particular, work investigating the mechanism of insulin mRNA cleavage by Ire1α, which has been implicated in type 2 diabetes.

proteins and cytoplasmic kinase/RNase domains, which are involved in the activation of downstream pathways. Mammals have two Ire1 paralogs, Ire1α and Ire1β. While Ire1α is a ubiquitously expressed gene, Ire1β expression is restricted to the gastrointestinal tract (*Tirasophon et al., 1998*; *Wang et al., 1998*). Upon ER stress, the Ire1α RNase domain is activated by self-oligomerization and subsequently excises a 26 base intron from the cytosolic unspliced form of XBP1u mRNA (*Yoshida et al., 2001*; *Calfon et al., 2002*). The resulting mRNA fragments are ligated in the cytosol by RtcB ligase generating the spliced form of XBP1 mRNA (*Jurkin et al., 2014*; *Kosmaczewski et al., 2014*; *Lu et al., 2014*). The Ire1α mediated splicing step is critical for mounting the UPR, as only the spliced XBP1 mRNA produces an active transcription factor that induces hundreds of genes responsible for increasing ER abundance to accommodate the demand for protein production (*Shaffer et al., 2004*; *Sriburi et al., 2004*; *Acosta-Alvear et al., 2007*). In addition to XBP1 mRNA splicing, Ire1α also reduces the load of incoming proteins by cleaving ER-localized mRNAs in a process termed regulated Ire1-dependent decay (RIDD) (*Hollien and Weissman, 2006*; *Han et al., 2009*; *Hollien et al., 2009*). While considerable attention has been paid to the mechanism of Ire1α activation, little is known about how the low abundant Ire1α efficiently finds and cleaves its mRNA substrates during ER stress. Recent studies have shown that XBP1u mRNA is recruited to the ER membrane through its nascent chain, but the components involved in the specific recruitment of XBP1u mRNA to the Ire1α cleavage site in the ER membrane remain unidentified.

In this study, we have discovered a complex between Ire1α and the Sec61 translocon channel in the ER membrane. We show that this interaction is specific by identifying key residues in Ire1α, and that it is stable even during ER stress conditions. Surprisingly, we find that a hydrophobic region in the XBP1u protein mimics a TMD and is co-translationally captured by the signal recognition particle (SRP). The SRP bound XBP1u-ribosome nascent chain (RNC) is then delivered to the Sec61 translocon

where its mRNA engages with Ire1α. Despite its interaction with the Sec61 translocon, the XBP1u nascent chain inefficiently inserts into the ER membrane due to its weak hydrophobic region. Furthermore, siRNA mediated depletion of SRP, the SRP receptor or the Sec61 translocon in human cells impairs the Ire1α-mediated splicing of XBP1u mRNA. Mutations in Ire1α that disrupts its association with the Sec61 translocon lead to reduced Ire1α-mediated cleavage of ER-targeted mRNAs. Over all, our studies establish an important link between the UPR and the co-translational protein translocation pathway, which ensures efficient cleavage of ER-targeted mRNAs during ER stress conditions.

## Results

### Ire1α is in a complex with the Sec61 translocon

To investigate how the low abundant Ire1α efficiently finds and cleaves its mRNA substrates, we searched for Ire1α associated proteins that could facilitate interactions with its mRNA substrates. To this end, we performed immunoaffinity purification from detergent solubilized microsomes derived from HEK 293 cells expressing hemagglutinin (HA)-tagged Ire1α. The affinity-purified material was subjected to SDS-polyacrylamide gel electrophoresis and bands not present in the control were subjected to analysis by mass spectrometry. Remarkably, in addition to a known Ire1α interacting protein, BiP (*Bertolotti et al., 2000*), we identified all three subunits of the Sec61 translocon (Sec61α, Sec61β, and Sec61γ) and Sec63 (*Figure 1A* and *Figure 1—figure supplement 1*). We were intrigued by the interaction between the Sec61 translocon and Ire1α since we reasoned that it could facilitate Ire1α access to RIDD mRNA substrates (*Hollien and Weissman, 2006*) that are targeted to the Sec61 translocon via their nascent chains. To determine if this interaction occurs at endogenous levels of Ire1α and the Sec61 translocon, we co-immunoprecipitated the endogenous Sec61 translocon from a detergent cell extract of non-transfected HEK 293 cells using Sec61β antibodies. We could detect a significant amount of endogenous Ire1α precipitating with Sec61β and Sec61α by immunoblotting (IB) (*Figure 1B*). Furthermore, immunodepletion of the Sec61 translocon nearly quantitatively depleted the endogenous Ire1α, indicating that almost all Ire1α is in a complex with the Sec61 translocon in cells (*Figure 1—figure supplement 2*). Interestingly, this interaction remained stable even after treatment of cells with the ER stress inducer DTT, which impairs protein folding by preventing disulfide bond formation in the ER lumen (*Figure 1B*). These results implied that Ire1α interaction with the Sec61 translocon might be functionally important during the conditions of ER stress. We further verified that the Sec61 translocon selectively associated with Ire1α, but not with the Ire1α paralogue Ire1β or with the other ER stress sensors PERK (*Harding et al., 1999*; *Sood et al., 2000*) and ATF6α (*Haze et al., 1999*) (*Figure 1C*). To determine whether the Ire1α interaction with the Sec61 translocon is direct, we treated HA-tagged Ire1α expressing cells with a lysine-reactive reversible crosslinker, DSP. After quenching the crosslinker, the complex was denatured in urea and SDS, which dissociates noncovalently bound proteins, and immunoprecipitated (IP) with HA antibodies. The resulting IP was treated with DTT to reverse the crosslinking and analyzed by IB. The α subunit of the Sec61 translocon could be crosslinked with Ire1α, as judged by the increase in Sec61α signal with increasing concentration of crosslinker (*Figure 1D*), thus supporting a direct interaction between Ire1α and the Sec61 translocon. Consistent with the result from Sec61β immunoprecipitation (*Figure 1C*), the crosslinked adduct was visible even when cells were treated with DTT before the crosslinker reaction, supporting a model where Ire1α associates with the Sec61 translocon even under ER stress conditions. (*Figure 1D*).

### A conserved region in Ire1α is required for the interaction with the Sec61 translocon

To exclude the possibility that this interaction was captured during Ire1α synthesis at the Sec61 translocon, we set out to identify specific residues in Ire1α that are required for the interaction with the Sec61 translocon. We therefore performed co-immunoprecipitation with HA antibodies using detergent extracts of cells expressing mutant versions of Ire1α-HA. While deletion of the luminal domain of Ire1α (amino acids 30 to 408) had no effect on the interaction with the Sec61 translocon, deletion of an evolutionarily conserved 10 amino acid region (amino acids 434 to 443) in the luminal portion of Ire1α adjacent to its TMD nearly abolished the interaction with the Sec61 translocon (*Figure 2A,C*). Mutagenesis of single residues within this region further revealed that

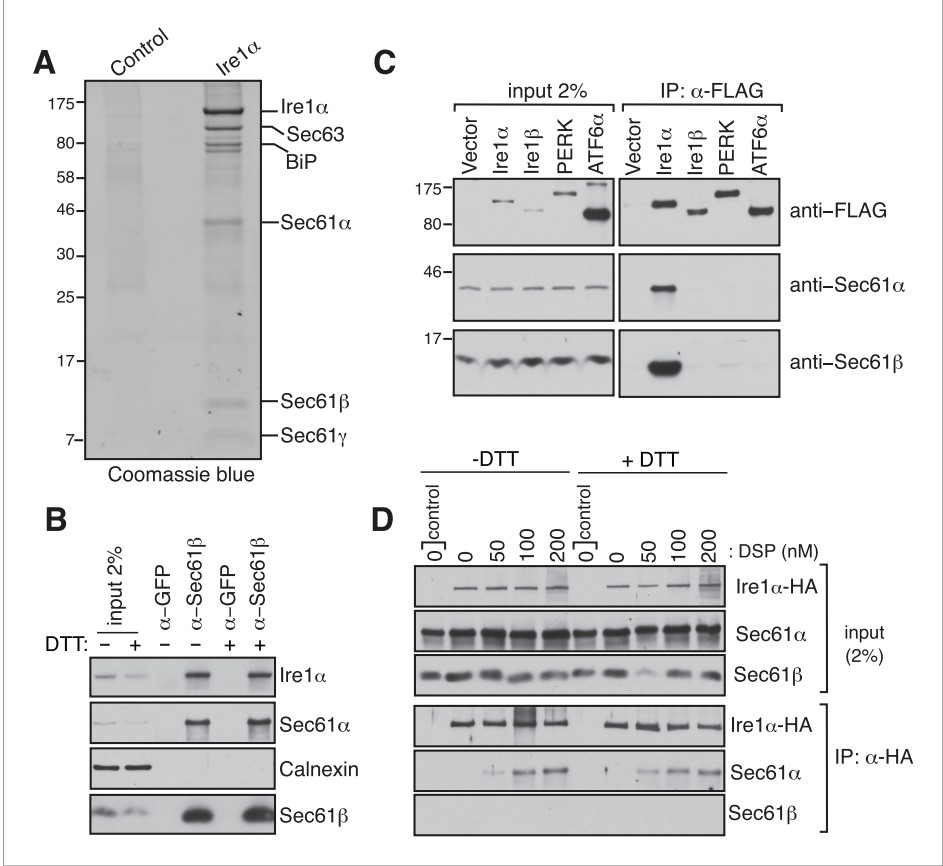

**Figure 1**. Identification of a complex between Ire1α and the Sec61 translocon. (**A**) The detergent extracts of either microsomes derived from HEK 293 cells (control) or cells expressing hemagglutinin (HA)-tagged Ire1α were bound to anti-HA resin and eluted with a low pH glycine buffer. The eluted proteins were analyzed by SDS-PAGE and stained with coommassie blue. (**B**) The cell lysates from non-transfected HEK 293 cells treated with or without DTT were immunoprecipitated (IP) with anti-GFP antibodies as a control or anti-Sec61β antibodies. The bound material was eluted with sample buffer and analyzed along with starting lysates (input, 2% loading) by immunoblotting (IB) using antibodies against the indicated antigens. Calnexin, an abundant endoplasmic reticulum (ER) trans membrane protein was probed as a control. (**C**) Cell extracts from HEK 293 cells transfected with the indicated FLAG tagged constructs were subject to IP with FLAG antibody. The resulting samples were analyzed by IB with indicated antibodies. (**D**) HEK 293 cells stably expressing HA-tagged Ire1α were either treated with 10 mM DTT or left untreated for 2 hr. Cells were then semipermeabilized with 0.015% digitonin and treated with the indicated concentration of DSP crosslinker for 30 min at room temperature. Samples were denatured and IP with anti-HA antibodies. The resulting IP was analyzed by IB. Control denotes non-transfected HEK293 cells.

The following figure supplements are available for figure 1:

**Figure supplement 1**. Peptides of Sec61α, Sec61β, and Sec61γ identified by mass spectrometry sequences of Sec61α, Sec61β, and Sec61γ annotated to indicate the peptides (yellow) identified by mass spectrometry.

**Figure supplement 2**. Ire1α is codepleted with the Sec61α translocon.

Val[437], Asp[438], Met[440], Leu[441] and Asp[443] are crucial for the interaction since replacing any of these amino acids with alanine significantly reduced the interaction with the translocon (*Figure 2B,C*). We next asked whether the Ire1α interaction with the Sec61 translocon is important for XBP1u mRNA cleavage during ER stress. To address this, we transiently depleted the Sec61α subunit, which forms the translocon channel, in cells by siRNA-mediated knock down. Indeed, Ire1α mediated splicing of XBP1u mRNA was substantially reduced in Sec61α depleted cells during ER stress (*Figure 2D*).

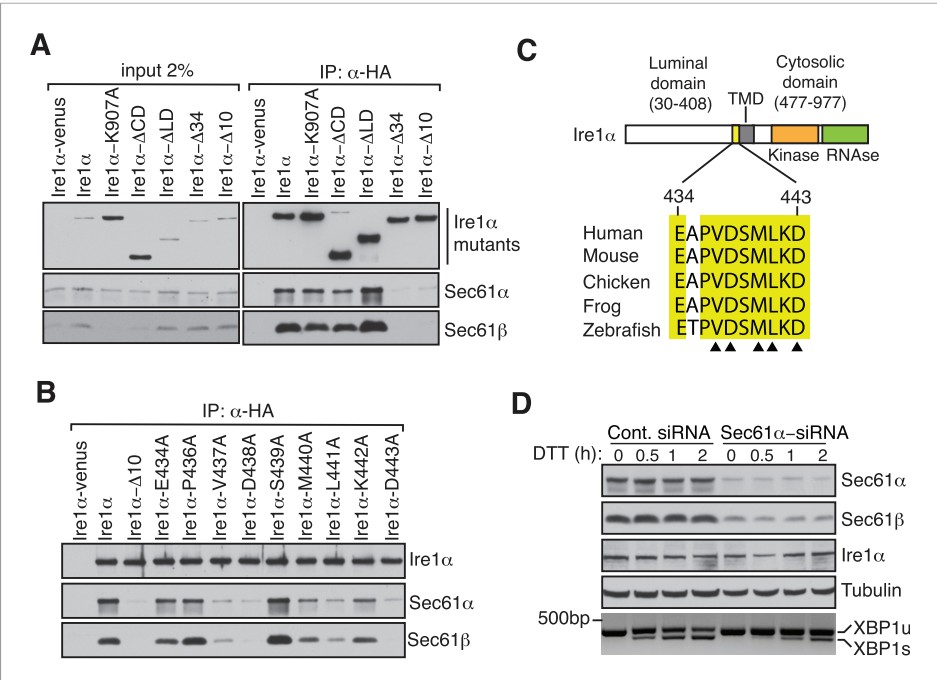

**Figure 2**. Key residues in Ire1α important for the interaction with the Sec61 translocon. (**A**) The cell lysates of the indicated versions of HA-tagged Ire1α were IP with anti-HA antibodies, eluted with sample buffer and analyzed by IB. Ire1α–venus served as a control. The mutation K907 to A907 impairs the RNase activity of Ire1α (**Tirasophon et al., 2000**). Deletion of the Ire1α cytosolic domain from amino acid 477 to 977 or luminal domain from amino acid 30 to 408 is labeled as ΔCD or ΔLD (**Volmer et al., 2013**), respectively. Ire1α Δ34 carry a deletion from amino acid 409 to 443 and Ire1α Δ10 lacks amino acids 434 to 443. (**B**) The indicated Ire1α mutants were analyzed as described in panel **A**. (**C**) Comparison of the sequences of the 10 amino acid region of Ire1α in vertebrates. Triangle depicts amino acid residues of Ire1α in which alanine scanning mutations disrupt binding to the Sec61 translocon. (**D**) HeLa cells were transfected with control siRNA or siRNA targeting Sec61α. After 48 hr of transfection, cells were transfected again with siRNA which was followed by transfection with FLAG-tagged XBP1u. 96 hr after the first transfection, cells were treated with 10 mM DTT for the indicated time periods. Total proteins and RNA were isolated from Trizol harvested cells and analyzed by IB against the indicated antigens and by an RT-PCR reaction to monitor splicing of XBP1u mRNA (**Calfon et al., 2002**), respectively.

## XBP1u mRNA utilizes the SRP pathway for targeting to the Sec61 translocon

Our findings of Ire1α in a complex with the Sec61 translocon raised the important question of how XBP1u mRNA would be recruited to this complex in order to be cleaved by Ire1α during ER stress. Although XBP1u mRNA encodes a soluble protein and shuttles between the cytosol and nucleus (**Yoshida et al., 2006**), recent studies have shown that XBP1u is co-translationally targeted to the ER membrane for efficient splicing of its mRNA by Ire1α (**Yanagitani et al., 2009**). This targeting reaction also depends on a C-terminal hydrophobic region 2 (HR2) as well as a translational pausing sequence located in the extreme C-terminus of XBP1u (**Yanagitani et al., 2009, 2011**). Combined, HR2 and the pausing sequence are speculated to facilitate direct interaction of RNCs of XBP1u with the ER lipid bilayer (**Yanagitani et al., 2009**). We reasoned that the interaction between XBP1u and lipids might not selectively direct its mRNA-RNCs to the ER membrane within the cell. We therefore hypothesized that XBP1u-RNCs may directly interact with the Sec61 translocon for its specific recruitment to the ER membrane as well as to engage with Ire1α. To first determine if the recruitment of XBP1u-RNCs requires any ER membrane factor(s), we reconstituted XBP1u recruitment to the ER membrane in vitro. XBP1u transcripts lacking a stop codon were translated using a rabbit reticulocyte lysate containing [35]S-labelled methionine and ER-derived rough microsomes (RM). As expected, the truncated XBP1u transcripts produced XBP1u-RNCs that were recruited to RM, as indicated by XBP1u

peptides in the pellet fraction (*Figure 3A,B*). However, a mild trypsin digestion of RM rendered them inactive for XBP1u recruitment. XBP1u recruitment to the ER membrane was dependent on the presence of HR2 since its deletion (XBP1ΔHR2) abolished the membrane recruitment, while replacing HR2 with a stronger hydrophobic TMD from the transferrin receptor (XBP1u-TR) restored XBP1u recruitment to RM (*Figure 3A,B* and *Figure 3—figure supplement 1*). We obtained similar results when we analyzed full-length versions of XBP1u, thus arguing against membrane recruitment due to artificial stalling of XBP1u at the ribosome (*Figure 3B*, bottom). These results supported our hypothesis that XBP1u-RNCs can be recruited to the ER membrane via an interaction with an ER membrane factor.

Based on the above observation and the presence of a hydrophobic region (HR2) (*Yanagitani et al., 2009*) in the XBP1u protein, we hypothesized that XBP1u mRNA-RNC targeting to the ER membrane may be mediated by the SRP pathway. To test this, we affinity-purified RNCs of XBP1u from in vitro translation reactions via an N-terminal FLAG-tag and analyzed for SRP recruitment by IB. Indeed, SRP was enriched in RNCs of XBP1u but not in RNCs of XBP1ΔHR2 (*Figure 3C*). As expected, XBP1u-TR exhibited slightly increased binding to SRP since it contains a genuine TMD (*Figure 3C* and *Figure 3—figure supplement 1*). We next wondered whether the binding of SRP to HR2 of XBP1u-RNCs is essential for targeting to the ER membrane. To address this, we translated XBP1u mRNA in vitro using a wheat germ extract that lacks SRP that is compatible with the mammalian SRP receptor

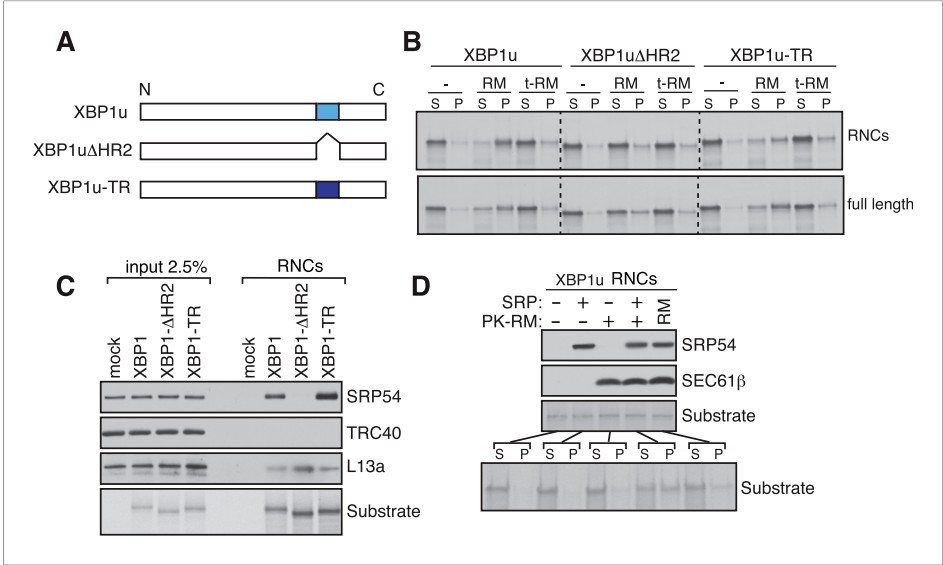

**Figure 3**. XBP1u utilizes the signal recognition particle (SRP) pathway for targeting its mRNA to the ER membrane. (**A**) Diagram of constructs derived from XBP1u. Blue box denotes the previously described hydrophobic region 2 (HR2) of XBP1u (*Yanagitani et al., 2009*). Dark blue indicates the transmembrane domain (TMD) from the transferrin receptor in lieu of HR2. (**B**) The indicated versions of XBP1u transcripts lacking a termination codon were translated in rabbit reticulocyte lysate in the presence of rough microsomes (RM) or trypsin digested RM (tRM). The reactions were separated by centrifugation to analyze pellet (P) and soluble fractions (S) by SDS-PAGE and autoradiography. (**C**) Affinity purified ribosome associated nascent chains (RNCs) of the indicated versions of FLAG-tagged XBP1u were analyzed by IB for the indicated antigens. L13a is a ribosomal protein. TRC40 is a control protein. Autoradiography of the blot revealed equal recovery of translated substrates. (**D**) XBP1u transcripts lacking a termination codon were translated in the wheat germ translation system including purified SRP, puromycin/ potassium acetate treated RM (PK-RM) or both. RM alone was included as a control. An aliquot of the total translation reaction was analyzed by IB for SRP54 and Sec61β, which indicate the presence of SRP and PK-RM, respectively. An autoradiograph of the blot revealed equal translation of substrate XBP1u in all reactions. The reactions were separated and analyzed as in panel **B**.

The following figure supplement is available for figure 3:

**Figure supplement 1**. Sequence and hydrophobicity of XBP1u constructs.

(*Walter and Blobel, 1981*). When RM stripped of ribosomes/SRP by puromycin and high salt (PK-RM) was added to this reaction, XBP1u nascent peptides were localized in the soluble fraction (*Figure 3D*). In contrast, adding back purified SRP to the original level found in RM shifted XBP1u localization to the pellet fraction, mirroring the membrane recruitment of XBP1u.

To examine whether Ire1α mediated splicing of XBP1u mRNA in cells also depends on the SRP mediated targeting of its RNCs to the ER membrane, we depleted SRP or the SRP receptor in cells by siRNA-mediated knock down. Cells depleted of the SRP subunits SRP14 or SRP54 showed sharply reduced XBP1u mRNA splicing upon treatment with the ER stress inducer DTT (*Figure 4A*). Similarly, depletion of the α subunit of the SRP receptor (SRα) nearly abolished splicing of XBP1u mRNA (*Figure 4B*). Importantly, these effects were not due to a defect in the biosynthesis of Ire1α since its level was unchanged by transient depletion of either SRP or its receptor (*Figure 4A,B*). To rule out the possibility that the reduction in XBP1u mRNA splicing observed in SRP knockdown experiments was not due to diminished ER substrate burden and Ire1α activation, we examined the activation of ER stress sensors under knockdown conditions (*Figure 4A,C*). Auto-phosphorylation of PERK and Ire1α in response to DTT treatment was identical under both control and SRP54/14 knockdown conditions (*Figure 4A*). In addition, depletion of SRP14, SRα or Sec61α had little effect on the amount of PERK phosphorylation and ATF6 cleavage in response to ER stress (*Figure 4C*). These data suggest that the reduced XBP1u mRNA splicing in the SRP pathway knockdown experiments is not an indirect effect but a result of reduced XBP1u mRNA targeting to the ER membrane. Together these results suggest that SRP binds to HR2 of XBP1u-RNC and recruits it to the ER membrane for Ire1α mediated splicing of XBP1u mRNA under ER stress conditions.

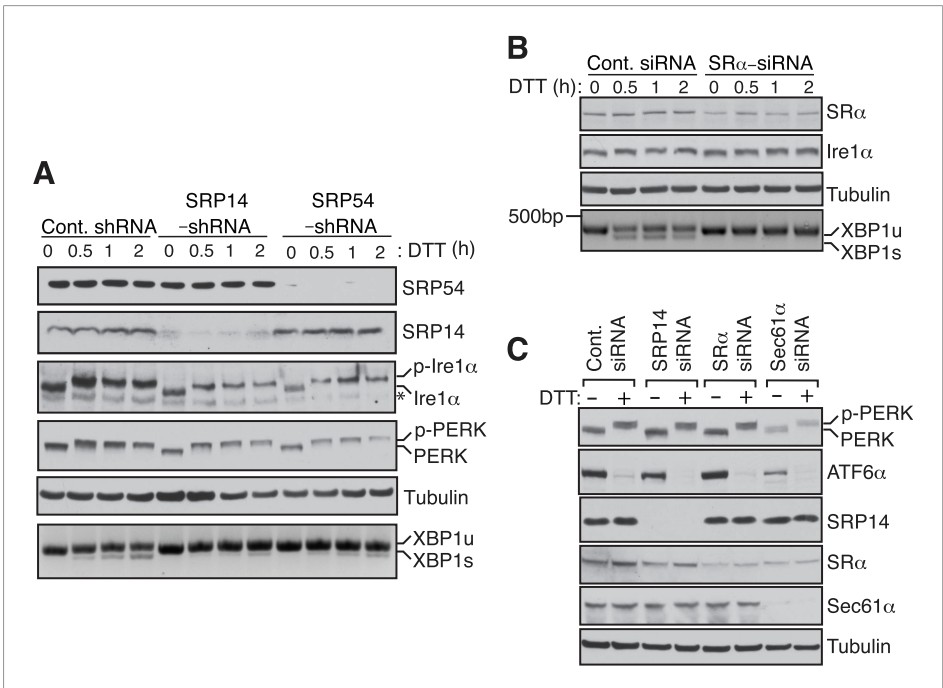

**Figure 4**. SRP mediated targeting to the ER ensures efficient spicing of XBP1u mRNA. (**A**) HEK 293 cells were transfected with shRNAs against luciferase (control), SRP14, or SRP54. 5 days after transfection, the cells were replated for transfection with XBP1u and treated with 10 mM DTT for the indicated time periods. The Trizol harvested cells were analyzed as in *Figure 2D*. A phos-tag gel was used for the Ire1α immunoblot. p-Ire1α and p-PERK indicate the phosphorylated forms of Ire1α and PERK, respectively. * denotes a background band. (**B**) HeLa cells were transfected with control siRNA or siRNA targeting the α subunit of the SRP receptor (SRα) and analyzed as described in *Figure 2D*. (**C**) HeLa cells were transfected with the indicated siRNA oligos and treated with 10 mM DTT for 2 hr after 96 hr post-transfection. Cells were harvested in SDS sample buffer and analyzed for IB with indicated antibodies. Upon DTT treatment, the ATF6α band disappears due to the cleavage of its N-terminal cytosolic domain. Our ATF6α antibodies were not suitable for detecting the cleaved N-terminal cytosolic domain (not shown). Note that depletion of Sec61α caused significant reduction of the transmembrane proteins PERK and ATF6α.

## XBP1u RNCs interact with the Sec61 translocon, but weakly insert into the ER membrane

Since the XBP1u protein does not possess a typical TMD, we wondered whether the SRP bound XBP1u-RNC is actually delivered to the Sec61 translocon in the ER membrane. To examine this, we isolated the ER membrane-targeted RNCs of XBP1u from in vitro translation reactions and treated them with a cysteine reactive chemical crosslinker. We observed a weak crosslinking between XBP1u and the Sec61α subunit (*Figure 5A*, lanes 2 and 4). By contrast, XBP1u strongly crosslinked to the β subunit of the translocon, presumably via a single cysteine residue localized in its cytosolic domain (*Figure 5A*, lanes 2 and 5). This result implied that the moderate hydrophobicity of HR2 of XBP1u might prevent full engagement of the translocon. This was further supported by the improved crosslinking between XBP1u and Sec61α observed when we used XBP1u-TR, which contains a genuine TMD (*Figure 5A*, compare 4 and 7). It has been previously shown that the Sec61 translocon can discriminate between functional and non-functional or weak signal sequences (*Jungnickel and Rapoport, 1995*). Accordingly, the functional signal sequence is able to form a protease resistant tight junction with the Sec61 translocon. To directly test if RNCs of XBP1u form a weak or tight junction with the Sec61 translocon, we performed a proteinase K (PK) accessibility assay with the ER membrane targeted RNCs of XBP1u. We noticed efficient protection of the XBP1u signal sequence HR2 under physiological salt concentration but it became partially PK sensitive under high salt conditions, suggesting that XBP1u HR2 forms a weak junction with the translocon (*Figure 5B* compare band 1 in lane 2 and 3). By contrast, we observed a tight complex between XBP1u-TR and the translocon as judged by an increased protection of XBP1-TR relative to XBP1u under high salt conditions (*Figure 5B* compare band 1 in lane 3 and 8). Interestingly, upon high salt and puromycin treatment, which releases nascent chains from ribosomes, we detected membrane-protected fragments for XBP1-TR that disappeared after treating with a detergent (*Figure 5B* lane 9, 10), demonstrating that insertion of XBP1u-TR occurs after its release from the ribosome. However, we failed to detect membrane-protected fragments for XBP1u, suggesting that XBP1u HR2 is rejected by the Sec61 translocon after its release from the ribosome. Interestingly, we noticed ribosome protected fragments even after puromycin treatment, indicating that the interaction between the translational pausing sequence of XBP1u and the ribosome exit tunnel remains stable (*Figure 5B* band 2 in lanes 5 and 10).

Since we observed an interaction between XBP1u and the translocon, we wondered whether this interaction facilitates integration of XBP1u into the ER membrane. To address this, we introduced an N-glycan acceptor site prior to the stop codon of XBP1u constructs and translated them in the presence of RM. Glycosylation, a post-translocational event diagnostic of successful insertion, was readily detected for XBP1u, as determined by endoglycosidase H (Endo H) deglycosylation (*Figure 5—figure supplement 1*). As expected, no glycosylation was detected for XBP1ΔHR2, whereas XBP1-TR showed increased glycosylation. Importantly, the co-translational protein insertion pathway was solely responsible for these insertion activities since almost no glycosylation was detected during a post-translational protein insertion assay (*Figure 5—figure supplement 1*). Furthermore, a time course experiment revealed a decreased rate of glycosylation for XBP1u relative to its counterpart XBP1u-TR, suggesting that its weak hydrophobic region HR2 impedes efficient insertion into the membrane (*Figure 5C*). Corroborating the in vitro results, in cells expressing XBP1u constructs we detected the HR2 dependent glycosylation of XBP1u, which was sensitive to the Endo H or peptide-N-glycosidase (PNGase) treatment (*Figure 5D*). As expected, the XBP1u glycosylation was significantly less efficient relative to that of XBP1u-TR (*Figure 5D*). These findings are consistent with recent observations that a fraction of XBP1u could be inserted into the ER membrane in cells (*Chen et al., 2014*). Collectively, these results suggest that the sequence of XBP1u HR2 has evolved in a way that it manages to follow the co-translational protein translocation pathway, but avoids efficient insertion into the ER membrane.

## Disrupting the complex between Ire1α and the Sec61 translocon reduces Ire1α-mediated cleavage of ER-targeted mRNAs

We next examined whether disrupting the interaction between Ire1α and Sec61 translocon impairs Ire1α mediated cleavage of XBP1u mRNA. To this end, we complemented Ire1α or the translocon interaction defective Ire1α mutants into either HEK 293 Ire1α$^{-/-}$ cells generated by the CRISPR/Cas9 system or mouse embryonic fibroblast (MEF) Ire1α$^{-/-}$ cells (*Lee et al., 2002*). The complementation of

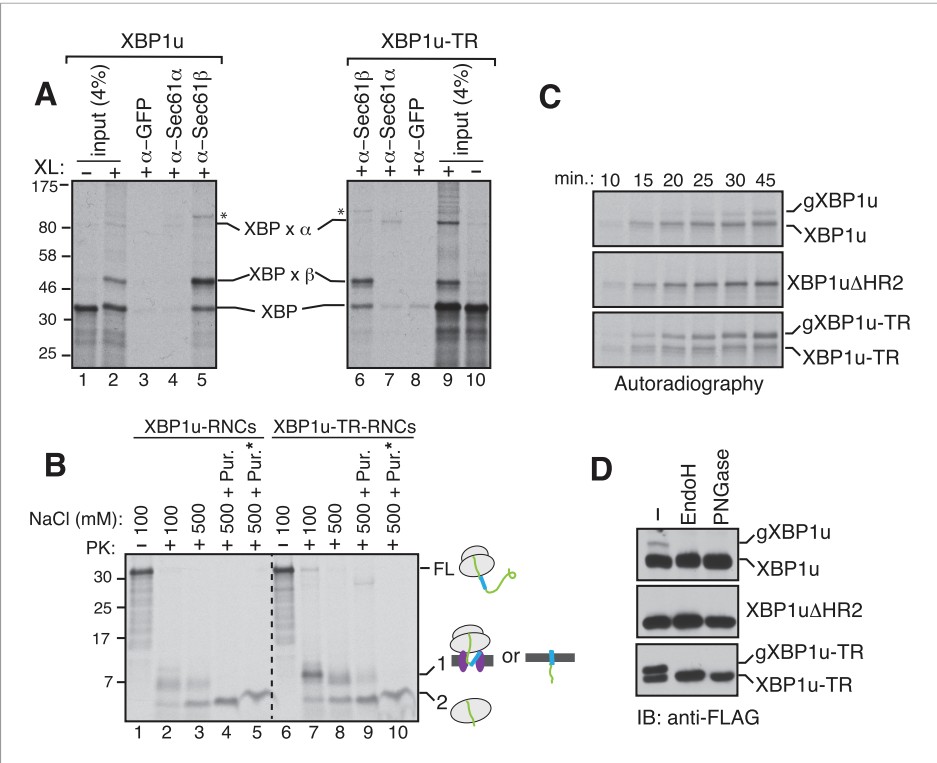

**Figure 5**. XBP1u nascent chains interact with the Sec61 translocon, but inefficiently insert into the ER membrane. (**A**) The membrane-targeted RNCs of XBP1u or XBP1u-TR were isolated by centrifugation and treated with BMH crosslinker. An aliquot was directly analyzed (input, 4% loading), while the remainder was IP with the indicated antibodies. Anti-GFP antibodies were used as a control. The XBP1u crosslinked adducts are indicated by 'XBP x'. * indicates an unidentified crosslinked product. (**B**) The ER membrane targeted RNCs of XBP1u variants were subjected to a proteinase K (PK) accessibility assay in the presence of the indicated salt concentrations and salt plus puromycin (pur.). * indicates the inclusion of a detergent in the reaction. FL indicates full-length versions of XBP1u. Band 1 indicates protease-protected fragments of either ribosome translocon protected fragments or protected fragments after insertion into the membrane (lane 9). Band 2 indicates fragments protected by ribosomes. (**C**) The indicated versions of XBP1u transcripts containing a glycan acceptor site at the C-terminus were translated in vitro in the presence of RM. The reactions were stopped at the indicated time points by directly mixing with the sample buffer and analyzed by autoradiography. gXBP1u denotes the glycosylated form. (**D**) Cell lysates from HEK 293 cells expressing the indicated FLAG tagged XBP1u versions containing a glycan acceptor site were treated with endoglycosidase H (EndoH) or peptide-N-glycosidase F (PNGase) and analyzed by IB with FLAG.

The following figure supplement is available for figure 5:

**Figure supplement 1**. Insertion assays with XBP1u and its variants.

Ire1α into Ire1α$^{-/-}$ cells led to restoration of XBP1u mRNA splicing in an ER stress dependent manner, whereas the complementation of Ire1α mutants either Δ10 or D443A showed sharply reduced XBP1u mRNA splicing (*Figure 6A,B*). In addition, the Ire1α mutant D443A also exhibited a significant deficiency in downregulation of the RIDD mRNA substrates Blos1 and Scara3 (*Hollien et al., 2009*) (*Figure 6C*). These effects were not due to a defect in activation of Ire1α mutants under ER stress conditions since we observed similar Ire1α auto-phosphorylation in both wild type and Δ10 Ire1α expressing cells (*Figure 6D*). These results support the model that the Sec61 translocon bridges Ire1α and its mRNA substrates (*Figure 7*).

## Discussion

In the present study, we have addressed how the low abundant Ire1α effectively finds and cleaves its substrate mRNAs that are associated with ribosomes in the ER membrane. Our results have

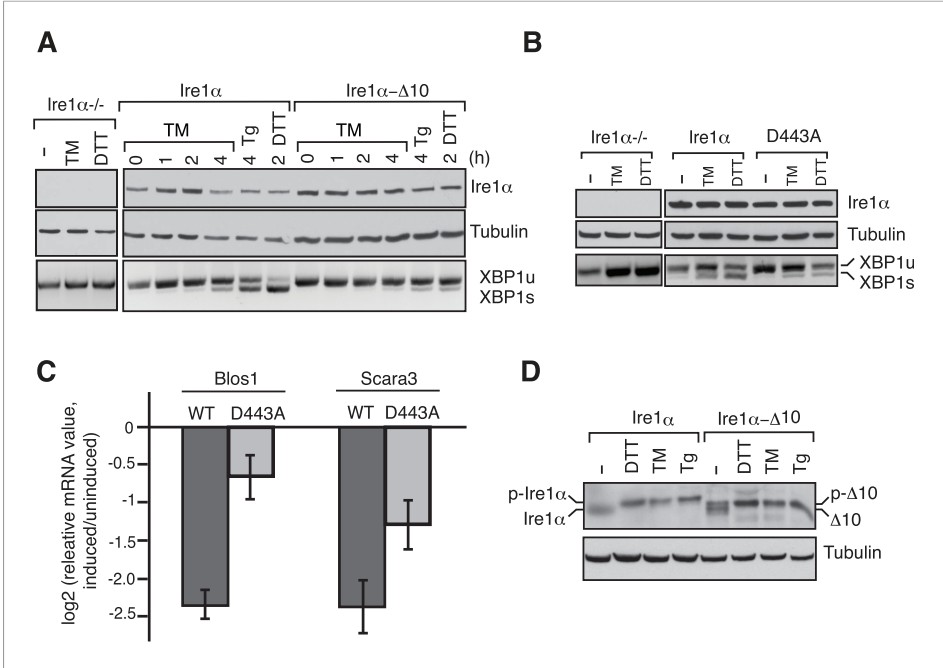

**Figure 6**. The Ire1α interaction with the Sec61 translocon ensures efficient cleavage of ER-targeted mRNAs. (**A**) HEK 293 Ire1α⁻/⁻ cells generated by CRISPR/Cas9 were stably complemented with Ire1α-HA or its mutant (Δ10). The expression of these constructs was controlled by doxycyline, but the cells were not induced with doxycyline in order to achieve low expression levels of Ire1α. Cells were harvested in Trizol after either treatment with tunicamycin (TM: 5 µg/ml), thapsigargin (Tg: 2.5 µg/ml) or DTT (10 mM) for the indicated time periods and analyzed by XBP1u mRNA splicing assay and IB with the indicated antibodies. (**B**) Mouse embryonic fibroblast (MEF) Ire1α⁻/⁻ cells complemented with Ire1α-HA or its mutant (D443A) were harvested after either treatment with TM (5 µg/ml) for 5 hr or DTT (10 mM) for 2 hr and analyzed by XBP1u mRNA splicing assay and IB as described in *Figure 2D*. (**C**) The MEF Ire1α⁻/⁻ cells complemented with indicated Ire1α variants were treated with TM (5 µg/ml) for 6 hr and analyzed by qPCR to measure Blos1 and Scara3 mRNA abundance. We normalized all mRNA abundance measurements to the housekeeping control Rpl19 mRNA. (**D**) HEK 293 Ire1α⁻/⁻ cells stably expressing Ire1α-HA or its mutant (Δ10) were treated with DTT for 2 hr, TM for 5 hr, Tg for 5 hr and analyzed for phosphorylated Ire1α.

established a direct link between the co-translational translocation pathway and the UPR that facilitates efficient cleavage of ER-targeted mRNAs by Ire1α during ER stress (*Figure 7*). Specifically, we have identified a complex comprising Ire1α and the Sec61 translocon, which is stable even during ER stress conditions. We have shown that this interaction is specific and is not captured while Ire1α is being synthesized in the Sec61 translocon since the other ER stress sensors, Ire1β, PERK or ATF6, fail to interact with the Sec61 translocon. Moreover, our domain mapping studies identified a conserved region in the luminal domain of Ire1α required for this interaction. Several observations suggest that Ire1α may directly interact or at least be in close proximity to the Sec61 translocon. First, our Ire1α pull down experiment identified the Sec61 translocon as one of the major interacting proteins in addition to Sec63 and BiP (*Figure 1A*). We can exclude the possibility that Ire1α associates with the Sec61 translocon through BiP since the interaction is stable even during ER stress conditions, whereas BiP dissociates from Ire1α (*Bertolotti et al., 2000*). Second, our chemical crosslinking studies captured a crosslinked adduct between Ire1α and the α subunit of the Sec61 translocon (*Figure 1D*). However, interaction studies with purified Ire1α and the Sec61 translocon are required to demonstrate if these proteins directly interact with each other. Interestingly, the Sec61 translocon interaction region is conserved only in vertebrates (*Figure 2C*), which suggests that the Ire1α interaction with the Sec61 translocon may not be absolutely essential for cleavage of its mRNA substrates but would increase the fidelity and efficiency of Ire1α.

We hypothesized that XBP1u mRNA might be targeted and localized in proximity to the Ire1α-Sec61 translocon complex in order to be efficiently cleaved during ER stress to yield an active

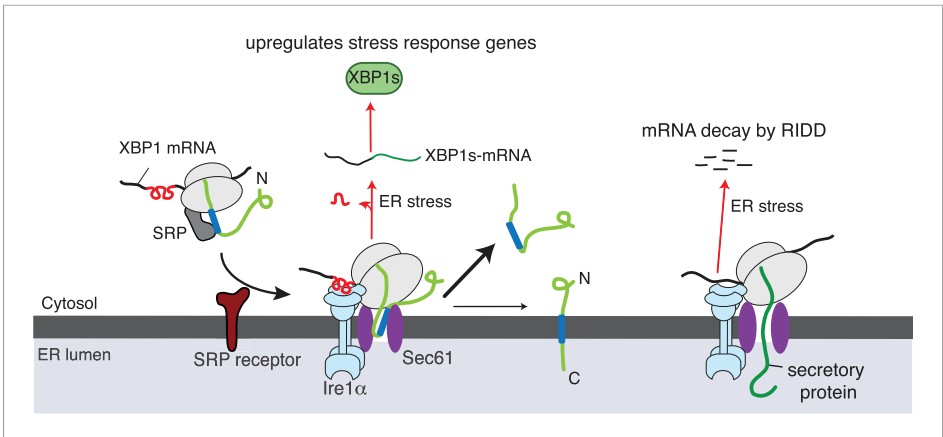

**Figure 7**. Model for Ire1α-mediated cleavage of ER-localized mRNAs. Ire1α forms a complex with the Sec61 translocon, to which XBP1u mRNA is recruited by its ribosome nascent chains (RNCs) through the SRP pathway. Despite interacting with the Sec61 translocon, the XBP1u nascent chain is inefficiently inserted into the ER membrane due to its weak hydrophobic region. Upon ER stress, Ire1α is activated through self-oligomerization and cleaves XBP1u mRNA to yield an active transcription factor, XBP1s, as well as to cleave ER-localized mRNAs through regulated Ire1-dependent decay (RIDD).

transcription factor for UPR target genes. Initial biochemical fractionation studies have shown that XBP1u mRNA is localized in the ER membrane although its encoded protein is soluble and localized both in the cytosol and nucleus (*Stephens et al., 2005*; *Yoshida et al., 2006*). Elegant studies from the Kohno group revealed that XBP1u mRNA is co-translationally targeted to the ER membrane as RNCs (*Yanagitani et al., 2009*). They further showed that this targeting reaction also depends on a hydrophobic region present in the C-terminus of XBP1u as well as a translational pausing sequence (*Yanagitani et al., 2011*). Based on the observation that the HR2 of XBP1u associates with protein-free liposomes, they concluded that XBP1u RNCs could directly bind with the ER lipid bilayer. Although these studies have established the importance of co-translational targeting of mRNA-XBP1 RNCs to the ER membrane, it was not clear how these nascent chains are specifically recruited to the ER membrane in the presence of other membrane compartments in the cell. In an attempt to address this issue, we reproduced the ER membrane recruitment assay with XBP1u RNCs in vitro. Consistent with the previous findings, XBP1u RNCs were robustly recruited to the ER membrane in an HR2 dependent manner (*Figure 3B*). However, XBP1u RNCs could not be efficiently recruited to ER membranes treated with trypsin, which suggested that an ER membrane factor might be required for binding with XBP1u RNCs (*Figure 3B*). The hydrophobic sequence HR2 dependent targeting of XBP1u RNCs to the ER membrane led us to propose that the SRP pathway might be involved in this targeting reaction. Our biochemical reconstitution studies have revealed that SRP can capture HR2 of XBP1u RNCs, which are then associated with the ER membrane by an interaction with the Sec61 translocon (*Figure 3C,D* and *Figure 5A*). It is possible that the weak hydrophobic sequence of HR2 of XBP1u RNCs can be captured in vitro but may not effectively compete for the SRP binding in cells due to the presence of numerous substrates of SRP, which are likely of stronger hydrophobicity than HR2 of XBP1u. However, our results argue against this idea since siRNA-mediated depletion of co-translational protein translocation components showed significantly reduced Ire1α-mediated cleavage of XBP1u mRNA in cells (*Figure 4A,B*). We speculate that the weaker signal sequence of HR2 might be strengthened by a translational pausing sequence in XBP1u, thus providing an increased time window for SRP binding. This is supported by the fact that co-translational pausing of XBP1u nascent chains are crucial for targeting to the ER membrane in vitro and in cells (*Yanagitani et al., 2011*).

We have shown that the SRP bound XBP1 RNCs are efficiently delivered to the Sec61 translocon, but do not form a tight complex with the Sec61 translocon (*Figure 5A,B*). Thus most are released into the cytosol and less than 10% are inserted into the ER membrane (*Figure 5C,D*). Future studies are required to precisely determine whether XBP1u nascent chains are released at the Sec61 translocon or after insertion into the ER lipid bilayer. Our results suggest that the moderate hydrophobicity of

XBP1u HR2 impedes efficient insertion into the ER membrane, since replacing it with a stronger TMD from the transferrin receptor significantly improved insertion into the ER membrane (*Figure 5C,D*). It is puzzling why XBP1u evolved with a combination of a weak C-terminus HR2 and a translational pausing sequence to utilize the SRP pathway rather than a stronger TMD or a signal sequence. In plants, Ire1 catalyzes the cytoplasmic splicing of bZIP60 mRNA to produce an active transcription factor (*Mishiba et al., 2013*). Interestingly, bZIP60 mRNA is likely targeted to the ER membrane by the SRP pathway since it encodes a typical transmembrane protein. Therefore, the specific advantage of this unique targeting mechanism utilized by XBP1u mRNA in vertebrates is unclear. It may be that the presence of a soluble form of XBP1u is necessary for a fully switchable UPR. This hypothesis is supported by earlier studies showing that the XBP1u protein binds to the active transcription factor XBP1s and routes it for proteasomal degradation (*Lee et al., 2003*; *Yoshida et al., 2006*). This feed back loop has been shown to be important for accurately turning off UPR genes by XBP1s when the ER stress is restored. Recent studies have shown that numerous cytosolic mRNAs are localized in the ER membrane through poorly understood mechanisms (*Reid and Nicchitta, 2015*). Our findings of the promiscuous substrate selectivity exhibited by SRP in binding moderate hydrophobic region in XBP1u suggests that other cytosolic proteins encoding mRNAs could be targeted to the ER membrane through the SRP pathway.

We hypothesize that one of the reasons Ire1α has evolved a specific interaction with the Sec61 translocon is to overcome the limitations imposed by its low abundance in the ER membrane relative to Sec61, where its substrate mRNAs are recruited (*Ghaemmaghami et al., 2003*). Indeed, our complementation experiments show that the Sec61 interaction defective Ire1α mutants were not able to efficiently mediate splicing of XBP1u mRNA during ER stress as well as cleavage of the RIDD mRNA substrates (*Figure 6A–C*). However, we found that overexpressing Ire1α mutants restores the inefficient cleavage of mRNAs (data not shown), suggesting that the Sec61 translocon interaction is important to bridge the low abundant Ire1α and its mRNA substrates. It is currently unclear how Ire1α is localized to the specific translocon where XBP1u mRNA may be found, though it is feasible that Ire1α may dynamically monitor the Sec61 translocon population and thus increase the likelihood of contact with translocon-localized XBP1u mRNA. Another intriguing hypothesis is that Ire1α may interact with a subset of the translocon population where XBP1u mRNA could be preferentially localized. In any case, it is likely that another layer of complexity may exist in order to facilitate efficient co-localization of Ire1α and XBP1u mRNA. In addition, since Ire1α interacts with the Sec61 translocon, both under normal and ER stress conditions (*Figure 1C,D*), it is unclear what prevents Ire1α from spuriously cleaving mRNAs associated with the Sec61 translocon under normal conditions. Most likely, accumulation of misfolded proteins triggers self-oligmerization and activation of the translocon-associated Ire1α only during ER stress conditions. How Ire1α is specifically arranged with the Sec61 translocon to access its substrate mRNAs and how it is coordinated with several other translocon-associated proteins remain important questions for future studies.

## Materials and methods

### DNA constructs

The in vitro expression Sp64 vector (Promega, Madison, WI) based construct encoding N-terminus FLAG-tagged XBP1u was generated from human XBP1u cDNA (Sino Biological, Inc. China) using standard molecular biology methods. XBP1uΔHR2 was created by deleting the amino acid coding sequence 186–208 using both 5′ phosphorylated oligos and the Phusion site directed mutagenesis protocol. XBP1u-TR was constructed by replacing HR2 (186–208) with the oligonucleotides encoding the TMD of human transferrin receptor (IAVIVFFLIGFMIGYLGY) by an overlap extension PCR method. The TMD of transferrin receptor serves as a signal sequence for recognition by SRP as previously described (*Mariappan et al., 2010*). We appended a 3F4-tag sequence (GTNMKHMAGAAA) to the C-terminus of XBP1u constructs, which end with asparagine amino acid (N), thus yielding an N-glycosylation motif (NGT). For the preparation of RNCs, the open reading frames were PCR amplified using a forward 5′ primer annealing to SP6 (*Sharma et al., 2010*), and a reverse primer lacking a stop codon.

For mammalian cell expression, we generated FLAG-tagged XBP1u with its 3′ UTR by following the previously described procedure (*Yanagitani et al., 2009*) and cloned into pcDNA5/FRT/TO (Invitrogen, Carlsbad, CA). By overlap extension PCR, a partial 3F4 tag sequence (GTNMKHM) was added prior to the stop codon of FLAG-XBP1u 3′ UTR in pcDNA5/FRT/TO, resulting in an NGT. The coding region of human

Ire1α was amplified from Ire1α-pcDNA3.EGFP (Addgene plasmid #13009, kindly provided by Dr Fumihiko Urano) and cloned into pcDNA5/FRT/TO carrying a C-terminal TEV protease cleavage site followed by either the HA- or the FLAG-tag. Similarly, the coding region of either mouse Ire1β or mouse PERK (Addgene plasmid #21880, #21814, kindly provided by Dr David Ron) was cloned into pcDNA5/FRT/TO carrying a C-terminal TEV protease cut site followed by the FLAG-tag. The human ATF6α was amplified including the 3xFLAG tag from p3xFLAG-ATF6 (Addgene plasmid #11975, kindly provided by Dr Ron Prywes) and cloned into pcDNA5/FRT/TO carrying a C-terminal TEV protease cut site followed by the HA-tag. The Ire1α (K907A) RNase mutant (*Tirasophon et al., 2000*) construct was made by site directed mutagenesis. Deletion of the Ire1α cytosolic domain (ΔCD) lacking amino acids 477–977, luminal domain (ΔLD) lacking amino acids 30–408 (*Volmer et al., 2013*), Δ34 lacking amino acids 409 to 443 and Δ10 lacking amino acids 434 to 443 were constructed using the Phusion site-directed mutagenesis protocol with the use of 5′ phosphorylated primers. Alanine scanning mutagenesis was performed using an efficient one step site directed mutagenesis protocol (*Zheng et al., 2004*). All PCR reactions were performed with Phusion high fidelity DNA polymerase (New England Biolabs, Ipswich, MA), except for site directed mutagenesis, which used Pfu-Ultra polymerase (Agilent Technologies, Santa Clara, CA). 3% DMSO was included in all PCR reactions to enhance amplification. The coding regions of all constructs were sequenced to preclude any sequence error.

## Antibodies and reagents

Antibodies were from the following sources: anti- L13a (Santa Cruz Biotech, Dallas, TX), anti-FLAG M2 antibody and anti-FLAG M2 affinity gel (Sigma–Aldrich, St. Louis, MO), anti-Ire1α (Cat. No. #3294, Cell Signalling, Beverly, MA), anti-PERK (Cat. No. #3192, Cell Signalling), anti-HA agarose (Cat. No. #11815016001, Roche, Switzerland) and complete protease inhibitor cocktail tablets (Roche), anti-Tubulin (Cat. No. #ab11312, Abcam, UK), anti-SRP54 (BD Biosciences, Franklin Lakes, NJ), anti-SRP14 (Cat. No. #PA5-27554, Fisher Scientific), Anti-HA (16B12, Cat. No. #MMS-101P-200, Covance, Princeton, NJ), anti-ATF6 (Cat. No. #sc-22799, Santa Cruz, Dallas, TX), Calnexin (Cat. No. #SPA-865, Enzo life sciences), Anti-mouse Goat HRP (Cat. No. #11-035-003, Jackson Immunoreserach, West Grove, PA), Anti-rabbit Goat HRP (Cat. No. #111-035-003, Jackson Immunoreserach) and Antibodies against TRC40, SRα, Sec61α, Sec61β, and GFP were previously described (*Snapp et al., 2004*). The purified SRP and the wheat germ translation system were purchased from tRNA probes, Texas. Reagents were from the following sources: Digitonin (EMD Millipore, Billerica, MA), DTT (American Bioanalytical, Natick, MA), Dithiobis (succinimidyl propionate) (DSP, Thermo Scientific, Waltham, MA) Bismaleimidohexane (BMH, Thermo Scientific) and protein-A agarose (RepliGen, Waltham, MA). siRNA oligos were purchased from Integrated DNA Technologies (San Jose, CA). Sec61α 3′ UTR siRNA (5′-CACUGAAAUGUCUACGUUUtt-3′) was previously described (*Lang et al., 2012*). SRα siRNA (5′-UAUAAACUGGACAACCAGUtt-3′) sequence was previously described, but it was used as an shRNA plasmid (*Lakkaraju et al., 2007*). shRNA plasmids of luciferase, SRP14, and SRP54 have been previously described (*Lakkaraju et al., 2007*).

## Cell culture and transfection

HeLa, HEK 293-Flp-In T-Rex (Invitrogen), and MEF *Ire1α*−/− FRT cell lines (*Hollien et al., 2009*) were cultured in high glucose DMEM containing 10% FBS at 5% $CO_2$. Transfections with either plasmids or siRNA oligos were performed with Lipofectamine 2000 (Invitrogen) according to the manufacture's protocol. For the siRNA mediated gene silencing, HeLa cells were transfected with 40 pmol of siRNA oligos per well in a 12 well plate using Lipofectamine 2000. At 24 hr after transfection, cells were replated for the second round of transfection with siRNA oligos at 48 hr. 6 hr later cells were transfected with 200 ng of pcDNA5-FLAG-XBP1u-3′ UTR. 96 hr after the first siRNA transfection, cells were harvested using the Trizol reagent (Invitrogen) for isolation of total proteins and RNA and analysed by IB and RT-PCR of XBP1-mRNA, respectively. shRNA mediated gene silencing in HEK 293 cells was performed by following the previously established protocol (*Lakkaraju et al., 2007*) except that at fifth day of transfection, cells were replated for transfection with FLAG-XBP1u-3′ UTR and harvested in Trizol after treatment with DTT at sixth day of initial transfection for analysis by both IB and RT-PCR of XBP1u mRNA. To establish stable cell lines, HEK 293-Flp-In T-Rex or MEF *Ire1α*−/− FRT cells were transfected with 1 μg of pOG44 vector (Invitrogen) and 0.1 μg of FRT vectors containing Ire1α or its mutants using Lipofectamine 2000. MEF *Ire1α*−/− FRT cells were plated in hygromycin (50 μg/ml) 24 hr after transfection, while HEK 293-Flp-In T-Rex cells were plated in

hygromycin (100 µg/ml) plus blasticidin (10 µg/ml). The medium was replaced every 3 days until colonies appeared. To regulate the expression of Ire1α or its mutants in complemented MEF Ire1α$^{-/-}$ cells, we transfected with pCDNA/tetR (Invitrogen, a kind gift from Dr Dhasakumar Navaratnam) and selected with blasticidin (10 µg/ml) until colonies appeared. The colonies were picked and the expression of recombinant Ire1α or its mutants was compared with control MEF cells.

## Immunoaffinity purification and co-immunoprecipitation

For purification of Ire1α associated proteins, HEK293 cells or HEK293 stable cells expressing Ire1α -HA were induced with 250 ng/ml of Doxycycline for 48 hr. Cells were lysed in Buffer A (10 mM Hepes pH 7.4, 250 mM Sucrose, 2 mM MgCl$_2$, 1× protease inhibitor cocktail) by repeated passage through a 23-gauge syringe needle. ER microsomes were isolated from low speed supernatants (2823×g for 30 min) by centrifugation for 1 hr at 75,000×g. Microsomes were resuspended in Buffer B (10 mM Hepes pH 7.4, 250 mM Sucrose, 2 mM MgCl$_2$, 0.5 mM DTT) and solubilized in lysis buffer (50 mM Hepes, 150 mM NaCl, 5 mM MgAc, 1 mM DTT, 1× protease inhibitor cocktail, 1% digitonin) for 30 min at 4°C. The supernatant was collected by centrifugation at 20,000×g for 15 min and incubated with rat anti-HA-agarose (Roche). The beads were extensively washed with lysis buffer, but containing 0.2% digitonin. The bound material was eluted from the column using 0.1 M glycine, pH 2.3 and 0.5% Triton-X100. The elutions were TCA precipitated and analysed by SDS-PAGE, followed by Coomassie blue stain. Bands of interest were identified by mass spectrometry at Keck MS and Proteomics Resource, Yale School of Medicine. For co-immunoprecipitation of endogenous Ire1α with the Sec61 translocon, HEK 293 cells were treated either with or without 10 mM DTT for 2 hr. Cells were lysed in Buffer A (50 mM Tris-HCl, pH 8.0, 150 mM NaCl, 1% digitonin) by rotating 30 min at 4°C. The supernatant was collected by centrifugation at 20,000×g for 15 min and incubated with anti-GFP or anti-Sec61β antibodies conjugated to protein-A agarose. The beads were extensively washed with Buffer A, but containing 0.2% digitonin. The bound material was eluted from the beads by directly boiling in SDS sample buffer. The clean blot IP system (Thermo Scientific) was used for secondary antibodies to minimize background from the primary antibodies. For co-immunoprecipitation of recombinant Ire1α and the endogenous Sec61 translocon, HEK 293 cells were transiently transfected with HA-tagged Ire1α versions. After 36 to 48 hr of transfection, cells were harvested in 1xPBS and lysed as above. The resulting digitonin cell extract was bound to anti-HA-agarose (Roche), washed, eluted SDS sample buffer and analysed by IB.

## Trypsin digestion of RM

RM derived from canine pancreas have been described (*Walter and Blobel, 1983*) RM was treated with 20 µg/ml of trypsin (Sigma) for 1 hr on ice. The reaction was stopped by adding 2 mM PMSF with continued incubation for 15 min on ice. The trypsin digested RM was sedimented through 0.5 M sucrose in a physiological salt buffer (PSB: 50 mM Hepes pH 7.4, 100 mM KAc, 2 mM MgAc) for 12 min at 70,000 rpm/TLA100.3 (Beckman, Brea, CA) to remove trypsin. The resulting pellet was resuspended in membrane buffer (50 mM Hespes pH 7.4, 250 mM Sucrose, 100 mM KAc, 2 mM MgAc, and 1 mM DTT). As a control, RM were treated similarly in parallel but without trypsin as a control.

## Membrane recruitment assay

This was done as described previously (*Yanagitani et al., 2011*) with the following modifications. Transcripts encoding versions of XBP1u lacking or containing a stop codon were translated in a rabbit reticulocyte lysate translation system (*Sharma et al., 2010*) supplemented with $^{35}$S-methionine in the presence or absence of membranes for 20 min at 32°C. The translation reaction was layered on 1 M sucrose prepared in PSB. After sedimentation for 15 min at 20,000×g, the supernatants and the pellets were analysed by SDS-PAGE and autoradiography.

## RNC affinity purification

RNCs of XBP1u versions were affinity purified using anti-FLAG agarose, similarly to previous methods (*Mariappan et al., 2010*). In brief, 300 µl reactions were translated for 20 min and immediately chilled on ice. The samples were adjusted to 2 mM cycloheximide, diluted to 1 ml with PSB, and incubated with 20 µl suspension of anti-FLAG affinity resin recognizing the N-terminal FLAG-tag for 1.5 hr. After being washed extensively with PSB, the bound RNCs were eluted in SDS sample buffer, and analysed by both IB and autoradiography.

## SRP dependent targeting in a wheat germ translation system

XBP1u transcripts lacking a termination codon were translated in a wheat germ extract supplemented with $^{35}$S-methionine, 32 nM purified SRP (tRNA probes) and/or puromycin/KAc treated RM (PK-RM). After incubation at 25°C for 45 min, the reactions were sedimented through 0.25 M sucrose in PSB for 15 min at 20,000×g. The supernatants and the pellets were analysed by SDS-PAGE and autoradiography.

## In vitro chemical crosslinking

Transcripts encoding versions of XBP1u lacking a termination codon were translated in the presence of RM for 25 min at 32°C. The membrane targeted RNCs were isolated by centrifugation through a 0.5 M sucrose cushion for 12 min at 70,000 rpm/TLA100.3, and the resulting pellet was resuspended in PSB. Crosslinking was performed with 400 µM BMH (a homo-bifunctional cysteine-reactive crosslinker) for 6 min at 25°C and quenched with 25 mM 2-mercaptoethanol. The resulting products were denatured with 1% SDS, 100 mM Tris-HCl pH 8.0 for 30 min at 55°C and diluted 10-fold IP buffer. The respective antibodies were added and incubated for 1.5 hr at 4°C, followed by incubation with protein-A agarose (Repligen) for 1.5 hr at 4°C. The beads were washed at least three times with IP buffer, eluted with SDS sample buffer and analysed after heating to 95°C, but 55°C/30 min for the Sec61α sample. Samples were treated with RNase A (100 µg/ml) for 10 min at 37°C before analyzing by SDS-PAGE and autoradiography.

## In vivo chemical crosslinking

This was done as described previously with the following modifications (Oyadomari et al., 2006). HEK 293 cells stably expressing HA-tagged Ire1α were semipermeabilized with 0.015% digitonin containing buffer (20 mM Hepes pH 7.4, 110 mM KAc, 2 mM MgAc) and treated with various concentrations of DSP crosslinker for 30 min at room temperature. Samples were collected in quenching/denaturing buffer containing 2% SDS, 6 M urea, 100 mM Tris-HCl pH 8.0, incubated for 30 min at 37°C, diluted 20-fold IP buffer and IP with anti-HA antibodies. The resulting IP was treated with DTT containing SDS sample buffer for 30 min at 37°C to reverse the crosslinking and analyzed by IB.

## Quantitative real-time PCR

For all RNA analyses, the total RNA was isolated from treated or non-treated cells using Trizol reagent (Invitrogen), treated with RQ1 RNase-Free DNase (Promega, Madison, WI) to remove residual DNA, and cDNA was synthesized using 2 µg of total RNA as a template, random hexamers, and M-MuLV reverse transcriptase (NEB). We measured relative mRNA abundance by real time quantitative PCR (BioRad, Hercules, CA) with SYBR green as the fluorescent dye. Each sample was measured in triplicate and normalized to Rpl19 mRNA levels. The primers for mouse Rpl19, Blos1 and Scar3 were previously described (Hollien et al., 2009).

## Ire1α auto-phosphorylation assay

A 6% polyacrylamide gel was made containing 25 µM Phos-tag (Wako, Japan) and 50 µM MnCl$_2$. SDS-PAGE was conducted at 100 V for 3 hr, followed by Mn chelation with 1 mM EDTA. The gel was transferred to nitrocellulose and probed with an anti-Ire1 antibody (Cell Signalling).

## CRISPR/Cas9 mediated depletion of Ire1α in HEK 293-Flp-In T-Rex

The Ire1α targeting sequence (5′ GATGGCAGCCTGTATACGCTTGG 3′) was cloned into the gRNA expression vector (Mali et al., 2013) in order to direct Cas9 nuclease activity toward the fourth coding exon of Ire1α. HEK 293-Flp-In T-Rex cells were plated in a six-well plate and transfected at 70% confluence with 250 ng of the gRNA expression vector and 250 ng of the pSpCas9(BB)-2A-Puro (Ran et al., 2013) expression plasmid with Lipofectamine 2000. Expression of Cas9 was selected for by puromycin treatment (2.5 µg/ml) for 48 hr, after which cells were returned to non-selecting media for 72 hr. Cells were then plated at 0.5 cells/well in 96 well plates and expanded for 3 weeks. Individual clones were examined for Ire1α knock out by IB against endogenous Ire1α.

## Acknowledgements

We are grateful to Ramanujan Hegde and Ajay Sharma for providing various antibodies and reagents. We thank Julie Hollien for MEF Ire1$^{-/-}$/FRT cells and Katharina Strub for shRNA plasmids. We thank Peter Cresswell, Ramanujan Hegde, and Christopher Burd for comments on this manuscript. We thank

Shawn Ferguson and Christian Schlieker for reagents and advice on CRISPR/Cas9, Matthew Simon for help with qPCR analysis and Jesse Reinhart for help with phos tag gels.

## Additional information

### Funding

| Funder | Grant reference | Author |
|---|---|---|
| Yale School of Medicine | Start-up Package | Rachel Plumb, Zai-Rong Zhang, Suhila Appathurai, Malaiyalam Mariappan |
| National Institutes of Health (NIH) | T32 GM007223 | Rachel Plumb |

The funders had no role in study design, data collection and interpretation, or the decision to submit the work for publication.

### Author contributions

RP, MM, Conception and design, Acquisition of data, Analysis and interpretation of data, Drafting or revising the article; Z-RZ, Conception and design, Acquisition of data, Analysis and interpretation of data; SA, Acquisition of data, Analysis and interpretation of data

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
