## [Decision Letter]

Thank you for sending your work entitled “A functional link between the co-translational protein translocation pathway and the UPR” for consideration at *eLife*. Your article has been favorably evaluated by Randy Schekman (Senior editor) and three reviewers, one of whom is a member of our Board of Reviewing Editors.

The Reviewing editor and the other reviewers discussed their comments before we reached this decision, and the Reviewing editor has assembled the following comments to help you prepare a revised submission.

Plum et al. describe an interaction between Ire1 and Sec61 that appears to be important for processing of mRNAs (XBP1u and RIDD substrates) by the Ire1 endonuclease. A hydrophobic segment in the XBP1-u nascent chain is responsible for targeting of XBP1u RNCs to the Sec61 complex. Depletion of subunits of the SRP, the SR or Sec61 reduces processing of XBP1u in cells that are treated with stress inducing agents. Plumb et al. have identified a short sequence in vertebrate Ire1α that is necessary for the Sec61-Ire1 interaction. While this manuscript builds on previous research indicating that XBP1u mRNA is translated in the vicinity of the RER, Plum et al. have outlined a reasonable mechanism to explain how a membrane bound endonuclease has access to a mRNA encoding a soluble protein. A revised version of this manuscript should be of interest to the readership of *eLife*. The most important points to address are listed below.

1) It is surprising to me that the authors do not emphasize pathway specificity of the effects in their paper, particularly with the SRP/SR/Sec61 knockdown experiments. Figure 4 is the only point where the PERK and ATF6 pathways are examined in this context, and they do not appear to be affected using the same stress conditions as in other figures (10 mM DTT for 2h). This is a very important point that merits emphasis because a major consequence of perturbing co-translational translocation is a markedly reduced load of ER clients. This may well blunt the effects of ER stressors whose actions rely on unfolded protein burden. Thus, a relatively trivial explanation for why knockdown of SRP/SR/Sec61 has reduced Xbp1 splicing is simply due to reduced substrate burden leading to reduced Ire1 activation. The strongest argument against this is direct illustration that in these same cells, activation of the other pathways is not affected. It therefore seems important to emphasize this in the paper, and ideally, have at least one carefully controlled experiment (e.g., Figure 2, Figure 4, and/or Figure 4) that simultaneously monitors both Ire1α and PERK activity to demonstrate selective effects on the former. Surprisingly I could not find experiments to test the effect of these depletions on IRE1 activation (assessed by its autophosphorylation status).

2) The evidence that the IRE1α-Sec61α interaction plays an important role in XBP1u splicing and the wholesale degradation of membrane bound mRNAs (so-called RIDD) is less compelling. It might be helpful to garner more evidence by comparing XBP1u mRNA splicing between wildtype IRE1α and a few other IRE1α mutants that fail to engage Sec61α in a complex when overexpressed (these are noted in Figure 2). It would also be helpful to know if these mutations have no effect on IRE1 activation, which would leave effects on substrate recruitment as their most plausible mode of action.

3) Information concerning the expression of Ire1 relative to Sec61 is necessary to understand the proposed mechanism. If Ire1 levels are low in abundance relative to Sec61 (e.g., <1/10) the interaction of Sec61 and Ire1 would have little impact on splicing. The authors should provide an estimate of the Sec61:Ire1 ratio in the HEK293 cells, which can probably be obtained from the data presented in Figure 1—figure supplement 2.

4) The authors need to clarify/revise the interpretation of the crosslinking experiment shown in Figure 5. XBP1u has three cysteine residues (C204, C215 and C247). C204 is missing from XBP1u-TR due to the exchange of the HR2 and TR-TM segment. C247 will be inside the large ribosomal subunit so C247 cannot crosslink to Sec61α or Sec61β. Canine Sec61β has a single cysteine residue. Given these facts, a ternary crosslinked product that contains XBP1u-TR, Sec61α and Sec61β is not possible using a maleimide crosslinker. Recovery of uncrosslinked XBP as a major band in the anti-Sec61β lanes casts doubt on the identification of the XBP1 x α x β band in Figure 5. It is also clear that this putative ternary product is not detected in the anti-Sec61α IP. This upper band is probably explained by crosslinking of C247 to a large ribosomal subunit protein that is located in the exit tunnel. Binary crosslinked products (XBP1u-Sec61β, XBP1u-TR-Sec61β and XBP1u-TR-Sec61α) are convincing, or very weak (XBP1u-Sec61α) consistent with the authors’ conclusion that XBP1u is targeted to the Sec61 translocation channel, but inefficiently inserted into the membrane.

---

## [Author Response]

*1) It is surprising to me that the authors do not emphasize pathway specificity of the effects in their paper, particularly with the SRP/SR/Sec61 knockdown experiments.*
Figure 4
*is the only point where the PERK and ATF6 pathways are examined in this context, and they do not appear to be affected using the same stress conditions as in other figures (10 mM DTT for 2h). This is a very important point that merits emphasis because a major consequence of perturbing co-translational translocation is a markedly reduced load of ER clients. This may well blunt the effects of ER stressors whose actions rely on unfolded protein burden. Thus, a relatively trivial explanation for why knockdown of SRP/SR/Sec61 has reduced Xbp1 splicing is simply due to reduced substrate burden leading to reduced Ire1 activation. The strongest argument against this is direct illustration that in these same cells, activation of the other pathways is not affected. It therefore seems important to emphasize this in the paper, and ideally, have at least one carefully controlled experiment (e.g.,*
Figure 2*,*
Figure 4*, and/or*
Figure 4*) that simultaneously monitors both Ire1α and PERK activity to demonstrate selective effects on the former. Surprisingly I could not find experiments to test the effect of these depletions on IRE1 activation (assessed by its autophosphorylation status)*.

We agree that this is an important point. We therefore repeated the experiment from Figure 4 and simultaneously monitored Ire1α and PERK activation as assessed by auto-phosphorylation. This data has now replaced the previous Figure 4 data. This data demonstrates that the activation of either Ire1α or PERK is not affected by depletion of SRP54 or SRP14. Indeed, while knockdown of co-translational translocation pathway components might be expected to decrease the ER protein folding load and thus reduce the ER stress response, this is not what we observed. Although it is unclear why this is the case, it may be that sufficient mRNAs are localized to the ER membrane during knock-down conditions to maintain a level of ER protein synthesis that enables the ER stress response. This idea is supported by the maintenance of Ire1α protein levels throughout the experiment, although the growth rate and cell numbers were reduced in SRP54 or SRP14 knock-down cells compared to control cells as indicated by the anti-tubulin western blot (Figure 4). Alternatively, it may be that the 10 mM DTT used to induce ER stress may not only prevent disulfide bond formation in newly synthesized proteins but may also break the disulfide bonds of existing proteins, thus inducing ER stress independently of the newly synthesized ER protein folding load. Finally, to emphasize the importance of this point, we have included a more in depth discussion of these experiments and their implications within the main body of the text.

*2) The evidence that the IRE1α-Sec61α interaction plays an important role in XBP1u splicing and the wholesale degradation of membrane bound mRNAs (so-called RIDD) is less compelling. It might be helpful to garner more evidence by comparing XBP1u mRNA splicing between wildtype IRE1α and a few other IRE1α mutants that fail to engage Sec61α in a complex when overexpressed (these are noted in*
Figure 2*). It would also be helpful to know if these mutations have no effect on IRE1 activation, which would leave effects on substrate recruitment as their most plausible mode of action*.

In order to gather more evidence for the role of the Ire1-Sec61α interaction in XBP1u mRNA splicing we established Ire1α ^*−/−*^ in HEK 293 cells using the CRISPR/Cas9 system. Consistent with the results from MEF Ire1α ^*−/−*^ cells, complementation of the Ire1α mutant ∆10 showed reduced XBP1u mRNA cleavage under ER stress compared to wild type Ire1α. This result further strengthens our model that the Sec61 translocon bridges Ire1α and its substrate mRNAs and has been included as Figure 6. Additionally, we examined activation of Ire1α, as determined by auto-phosphorylation, in HEK 293 Ire1α ^*−/−*^ cells complemented with either wild type or Ire1α-∆10 and treated with thapsigargin, tunicamycin, and DTT. We observed similar Ire1α auto-phosphorylation in both wild type and Ire1α-∆10 expressing cells under ER stress conditions (Figure 6), albeit a weak auto-phosphorylation was detected for ∆10 even under non-stress conditions.

*3) Information concerning the expression of Ire1 relative to Sec61 is necessary to understand the proposed mechanism. If Ire1 levels are low in abundance relative to Sec61 (e.g., <1/10) the interaction of Sec61 and Ire1 would have little impact on splicing. The authors should provide an estimate of the Sec61:Ire1 ratio in the HEK293 cells, which can probably be obtained from the data presented in*
Figure 1—figure supplement 2.

As the reviewers point out, this is an important issue for understanding the proposed mechanism of XBP1 splicing, but we are not sure how to obtain an estimate of the Sec61: Ire1α ratio from Figure 1—figure supplement 2. However, we quantified Ire1α concentration (∼1.4 nM) in either microsomes derived from HEK 293 cells or canine pancreatic microsomes using a quantitative western blot. We also quantified the Sec61 concentration (∼424 nM) in microsomes of HEK 293 by comparing to the previously estimated concentration (2.12 uM) of Sec61 in canine microsomes by Tyedmers et al. (2000, PNAS). We therefore estimate that Ire1α is ∼300 times less than the Sec61 in HEK 293 microsomes. It is also worth mentioning that although Ire1α and the Sec61 translocon form a tight complex, Ire1α was not detected in the previous biochemical purification of the Sec61 translocon (*Görlich et al. (*Cell, 1993), presumably because of its very low abundance in the ER membrane. Although our estimation is close to the ratio of the Sec61:Ire1 (96:1) quantified in yeast by Ghaemmaghami et al. (Nature, 2003), we did not to include this data in the manuscript since it is an indirect estimation. The purified Sec61 will be required to determine the precise ratio of Sec61: Ire1α in HEK 293 cells.

Author response image 1.Ire1α concentration in HEK 293 microsomes.Ire1α concentration in either HEK 293 microsomes or canine microsomes (CRM) was quantified relative to purified His-tagged Ire1α. We estimate 1μl of HEK microsomes (OD280 = 50) contain ∼150 pg or 1.4 nM of Ire1α. Tyedmers et al. (2000, PNAS) estimated that 1ul of CRM (OD280 = 50) contains ∼ 2.12 uM. Thus, 1 μl of HEK microsomes contain ∼ 0.4 uM since they contain 5X less concentration of Sec61. We therefore estimate that Ire1α concentration is ∼ 300X less than the Sec61 in HEK microsomes. Note that the five times less concentration of Sec61 in HEK relative to CRM is not likely biased by our Sec61α antibodies since both human and canine Sec61α peptide sequence are nearly identical.**DOI:**
http://dx.doi.org/10.7554/eLife.07426.016

*4) The authors need to clarify/revise the interpretation of the crosslinking experiment shown in*
Figure 5*. XBP1u has three cysteine residues (C204, C215 and C247). C204 is missing from XBP1u-TR due to the exchange of the HR2 and TR-TM segment. C247 will be inside the large ribosomal subunit so C247 cannot crosslink to Sec61α or Sec61β. Canine Sec61β has a single cysteine residue. Given these facts, a ternary crosslinked product that contains XBP1u-TR, Sec61α and Sec61β is not possible using a maleimide crosslinker. Recovery of uncrosslinked XBP as a major band in the anti-Sec61β lanes casts doubt on the identification of the XBP1 x α x β band in*
Figure 5*. It is also clear that this putative ternary product is not detected in the anti-Sec61α IP. This upper band is probably explained by crosslinking of C247 to a large ribosomal subunit protein that is located in the exit tunnel. Binary crosslinked products (XBP1u-Sec61β, XBP1u-TR-Sec61β and XBP1u-TR-Sec61α) are convincing, or very weak (XBP1u-Sec61α) consistent with the authors’ conclusion that XBP1u is targeted to the Sec61 translocation channel, but inefficiently inserted into the membrane*.

We appreciate reviewers for pointing out that there are not enough available cysteine residues for XBP1-TR to be cross-linked individually to Sec61α and Sec61β at the same time, since one of the two residues is most likely buried within the ribosome channel. We have now removed our conclusions about the ternary complex from the manuscript text and edited Figure 5 and legend appropriately.